# The gut efflux pump MRP-1 exports oxidized glutathione as a danger signal that stimulates behavioral immunity and aversive learning

Jonathan Lalsiamthara[1] & Alejandro Aballay [1✉]

Innate immune surveillance, which monitors the presence of potentially harmful micro-organisms and the perturbations of host physiology that occur in response to infections, is critical to distinguish pathogens from beneficial microbes. Here, we show that multidrug resistance-associated protein-1 (MRP-1) functions in the basolateral membrane of intestinal cells to transport byproducts of cellular redox reactions to control both molecular and behavioral immunity in *Caenorhabditis elegans*. *Pseudomonas aeruginosa* infection disrupts glutathione homeostasis, leading to the excess production of the MRP-1 substrate, oxidized glutathione (GSSG). Extracellular GSSG triggers pathogen avoidance behavior and primes naïve *C. elegans* to induce aversive learning behavior via neural NMDA class glutamate receptor-1 (NMR-1). Our results indicate that MRP-1 transports GSSG, which acts as a danger signal capable of warning *C. elegans* of changes in intestinal homeostasis, thereby initiating a gut neural signal that elicits an appropriate host defense response.

[1] Department of Molecular Microbiology and Immunology, School of Medicine, Oregon Health & Science University Portland, Oregon, OR 97239, USA.
✉email: aballay@ohsu.edu

The innate immune recognition of potential pathogenic microorganisms comprises the recognition of microbial-associated molecular patterns (MAMPs) and endogenous signals or damage-associated molecular patterns (DAMPs), which derive from cellular damage caused by infections. Unlike harmless commensal bacteria, microbial pathogens release an array of toxins or virulence factors that can alter cellular physiology and cause the release of DAMPs[1], whose detection is instrumental in activating innate microbe-killing mechanisms. To survive a pathogen assault, the host must also activate compensatory mechanisms to repair disabled cellular processes and detoxification programs. Thus, the simple act of toxic-substance removal from cells to reduce the exposure dose and duration may both improve cellular homeostasis and activate immune surveillance by releasing potential DAMPs.

Multidrug resistance-associated proteins (MRPs), members of the C-type family of ATP-binding cassette (ABC) transporters, can release a wide variety of endogenous or exogenous xenobiotic molecules across the cell membrane[2,3]. MRPs and MRP-like proteins are highly conserved and have been identified in all eukaryotes[3–5], for example, eight MRPs (MRP-1–8) are expressed in the nematode *Caenorhabditis elegans*[6]. In humans, alterations in the normal functioning of these efflux pumps are implicated in the etiology of diseases such as Dubin–Johnson syndrome[7] and cystic fibrosis[8–10]. MRP-1 has been implicated in the etiology of a wide array of human pathologies, including inflammatory bowel disease, cystic fibrosis, Alzheimer's disease, age-related macular degeneration, cardiovascular disease, and certain neurological disorders as well as tumor progression[10–17].

These broad roles of MRP-1 in human health and disease are thought to be due to its ability to mediate the efflux of bioactive organic anions and regulate oxidative stress[13,18–20]. Specifically, MRP-1 plays a major role in the transport and regulation of the most abundant (0.1–10 mM) low-molecular-weight antioxidant peptide[3,21], glutathione (GSH). MRP-1 exports out of cells reduced GSH with low affinity and its oxidized form, GSSG, with a higher affinity[20]. During oxidative stress, the level of GSSG increases within cells, and the efficient removal of GSSG molecules, which show pro-oxidant activity, is important for maintaining cellular health and redox homeostasis[3]. While the beneficial effects of actively removing toxic by-products of cellular insults, such as GSSG, via efflux pumps, have been well studied, the subsequent effects of these released by-products on neighboring cells and tissues are largely unknown.

GSSG, exported via MRP-1, may be recognized as a DAMP and result in the activation of defense responses. Indeed, increasing evidence indicates that the sensing of changes in core cellular processes or overall animal physiology can induce both molecular and behavioral innate immune responses. For example, in mammals, olfactory chemosensory neurons and nociceptor sensory neurons detect various bacterial products through distinct molecular mechanisms that lead to rapid avoidance behaviors[22–24]. In the fruit fly *Drosophila melanogaster*, olfactory and gustatory neurons have been reported to detect geosmin, phenol, and lipopolysaccharides, thereby allowing the organism to avoid foods contaminated with bacteria[25,26]. In *C. elegans*, behavioral pathogen avoidance is a crucial defense response that significantly improves the survival of animals[27–32].

In this study, we describe the role of MRP-1, a 171-kDa protein expressed in *C. elegans* with a high degree of homology to the human MRP-1 protein[33], and glutathione homeostasis in defense against pathogen infection. We observed that *C. elegans* mrp-1 mutants were highly susceptible to *Pseudomonas aeruginosa* infection, primarily due to defective pathogen avoidance. We found that *P. aeruginosa* infection induced a redox imbalance and drove the equilibrium toward excess GSSG production.

Extracellular supplementation with GSSG during infection resulted in enhanced avoidance behavior. Priming with GSSG molecules to mimic elevated systemic GSSG levels conditioned naive animals to avoid *P. aeruginosa*, thus inducing aversive learning in an NMDA-class glutamate-receptor-1 (NMR-1)-dependent manner. Rescue of MRP-1 in the intestine was sufficient to restore proper avoidance behavior as well as resistance to infection, indicating that MRP-1 functioned in the gut where it communicates with the nervous system in order to control pathogen avoidance. Our results indicate that the location of MRP-1 at the basolateral membrane of intestinal cells may facilitate the transport of GSSG directly into the pseudocoelomic cavity, where it can interact with nerve endings. We propose that by-products of cellular stress responses can be exported via specific efflux pumps and serve as signals that alert the nervous system to control behavioral immune responses.

## Results

**The MRP-1 efflux pump is required for *C. elegans* defense against *P. aeruginosa*.** To study the role of MRPs in the defense response against bacterial infections, we employed the model organism *C. elegans*, which is a genetically tractable nematode and convenient host, to study microbial pathogenesis and host responses to pathogenic infection[34,35]. We determined the susceptibility of *C. elegans* strains carrying mutations in MRPs to the human opportunistic pathogen *P. aeruginosa* strain PA14, a clinical isolate that is capable of rapidly killing *C. elegans* at 25 °C[36]. We found that *mrp-1(pk89)* animals were more susceptible to the pathogen than wild-type animals (Fig. 1a and Supplementary Fig. 1). The enhanced *P. aeruginosa* susceptibility of *mrp-1(pk89)* animals was also phenocopied by the RNA-interference (RNAi)-mediated knockdown of the *mrp-1* gene (Fig. 1b).

Two distinct mechanisms involved in the *P. aeruginosa*-mediated killing of *C. elegans* have been identified[36,37]. When *P. aeruginosa* is grown on low-nutrient media, *C. elegans* killing occurs over the course of several days, which is referred to as "slow killing." In contrast, when *P. aeruginosa* is grown on high-osmolarity media, *C. elegans* killing occurs over the course of several hours, which is referred to as "fast killing"[38]. Slow killing requires live bacteria and is correlated with the accumulation of *P. aeruginosa* in the *C. elegans* gut[36], whereas fast killing is mediated at least in part by low-molecular-weight toxins, including phenazines, and does not require live bacteria[37]. Given the ability of MRP-1 to export diverse compounds, we hypothesized that failure to export diffusible *P. aeruginosa* toxins, such as 1-hydroxyphenazine, phenazine-1-carboxylic acid, and pyocyanin, could play a major role in the *mrp-1* mutant phenotype. Under fast-killing conditions, in which diffusible toxins are the major determinants of lethal effects, *mrp-1(pk89)* animals exhibit a slightly greater susceptibility to *P. aeruginosa* than wild-type animals (Supplementary Fig. 2a). However, under these same conditions, *mrp-1(pk89)* animals were also more susceptible to a *P. aeruginosa* phenazine-deficient mutant (Δphz) than wild-type animals (Supplementary Fig. 2a). Under slow-killing conditions, in which phenazines only play a minor role, we found that *mrp-1(pk89)* animals exhibited enhanced susceptibility to *P. aeruginosa* Δphz relative to wild-type animals (Supplementary Fig. 2b). Taken together, these findings suggest that the susceptibility of *mrp-1(pk89)* animals to *P. aeruginosa*-mediated killing is not due to phenazines or diffusible toxins.

*C. elegans* naturally exhibits avoidance behavior when exposed to *P. aeruginosa*, and this phenomenon can be observed under conditions in which the animals can freely enter and exit a bacterial lawn[28,32,39,40]. On a full pathogen lawn, the survival

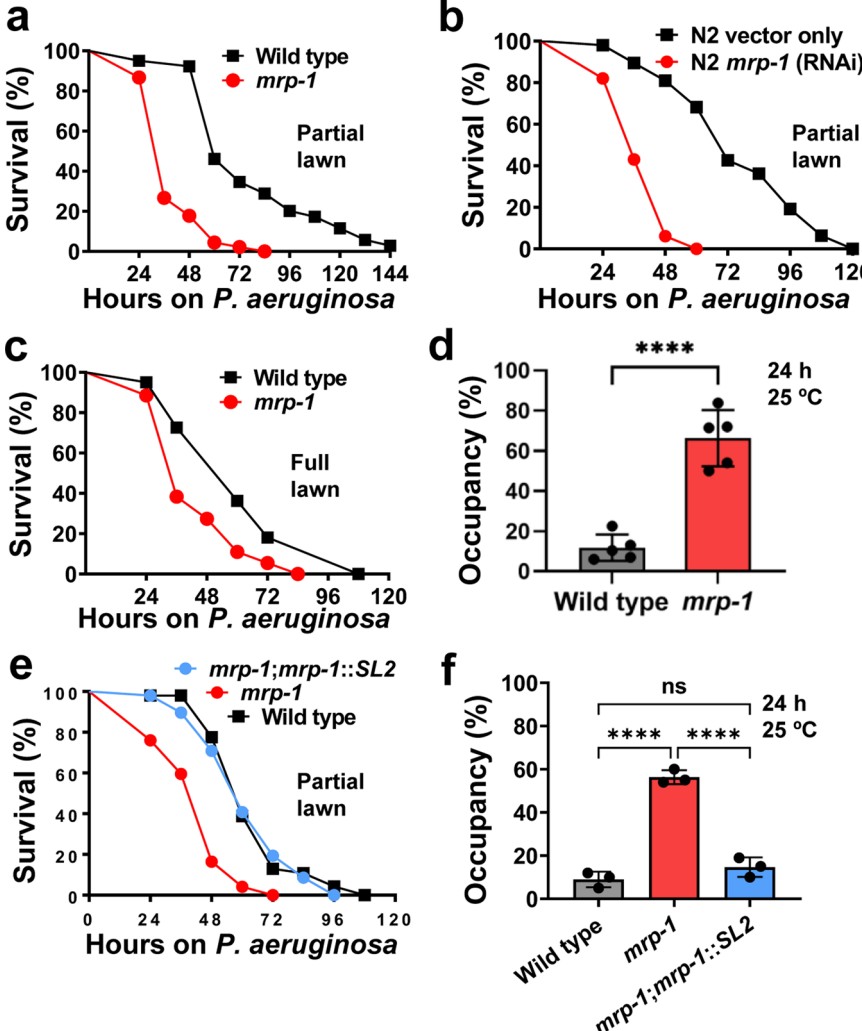

**Fig. 1 The MRP-1 efflux pump is required for resistance against pathogen infection. a** Representative survival plot of *C. elegans* wild-type and *mrp-1(pk89)* mutant animals exposed to a partial lawn of *P. aeruginosa*. The survival rate of *mrp-1* mutant animals was compared to that of wild type using the log-rank (Mantel–Cox) test, and the differences were found significant, ****$P < 0.0001$. **b** Representative survival plot of wild-type N2 vector only and *mrp-1* RNAi animals exposed to a partial lawn of *P. aeruginosa*. The survival rate of *mrp-1* RNAi animals was compared to that of wild type using the log-rank (Mantel–Cox) test, and the differences were found significant, ****$P < 0.0001$. **c** Representative survival plot of wild-type *C. elegans* and the *mrp-1(pk89)* mutant exposed to a full lawn of *P. aeruginosa*. The survival rate of *mrp-1* mutant animals was compared to that of wild type using the log-rank (Mantel–Cox) test, and the differences were found significant, **$P = 0.007$. **d** Lawn occupancy of wild-type *C. elegans* and the *mrp-1(pk89)* mutant on a partial lawn of *P. aeruginosa* at 24 h. The bars represent the mean ± SD from five independent experiments. ****$P ≤ 0.0001$ according to a two-tailed t-test. **e** Representative survival plot of wild-type, *mrp-1(pk89)* mutant, and MRP-1-supplemented transgenic animals exposed to a partial lawn of *P. aeruginosa*. The survival rate of *mrp-1;*MRP-1 transgenic animals was compared to that of wild type using the log-rank (Mantel–Cox) test, and the differences were found nonsignificant, ns $P = 0.6304$. **f** Lawn occupancy of wild-type, *mrp-1(pk89)* mutant, and MRP-1-supplemented transgenic *C. elegans* animals on a partial lawn of *P. aeruginosa* at 24 h. The bars represent the mean ± SD from three independent experiments. **** $P ≤ 0.0001$ according to one-way ANOVA with Tukey's multiple-comparison test. All assays were performed at 25 °C, with 40–50 animals/assay (**a–f**). 'ns' indicates nonsignificant; '*' indicates a significant difference.

difference between *mrp-1(pk89)* and wild-type animals was not as large as that observed on partial lawns (comparison of Fig. 1a, c). Because the assay using a full lawn eliminated the contribution of avoidance behavior to resistance against pathogen infection, these results indicated that MRP-1 may play a role in pathogen avoidance. Hence, we next directly investigated whether MRP-1 is involved in the avoidance response to *P. aeruginosa* by performing a lawn-occupancy assay, in which we calculated the fraction of animals on the bacterial lawn 24 h after they were placed on *P. aeruginosa*. The *mrp-1(pk89)* mutant animals showed a higher percentage of lawn occupancy than the wild-type animals (Fig. 1d), indicating that *mrp-1* is required for pathogen avoidance by *C. elegans*. We also examined the

avoidance behavior of *mrp-1(pk89)* mutant animals toward other bacterial species known to cause avoidance of *C. elegans*. We found that the lawn occupancy of *mrp-1(pk89)* mutant animals was comparable to that of wild-type animals on lawns of *E. faecalis* and *S. aureus* (Supplementary Fig. 3). This finding suggests that *mrp-1*-mediated behavioral control of pathogen avoidance is not universal and may be specific to *P. aeruginosa*.

To further support our findings, we rescued the activity of MRP-1 in *mrp-1(pk89)* animals using an extrachromosomal gene construct. MRP-1 expressed under its own promoter could reverse the susceptibility of *mrp-1(pk89)* animals to *P. aeruginosa* (Fig. 1e), as well as the avoidance defect of the mutant animals (Fig. 1f). Taken together, these findings

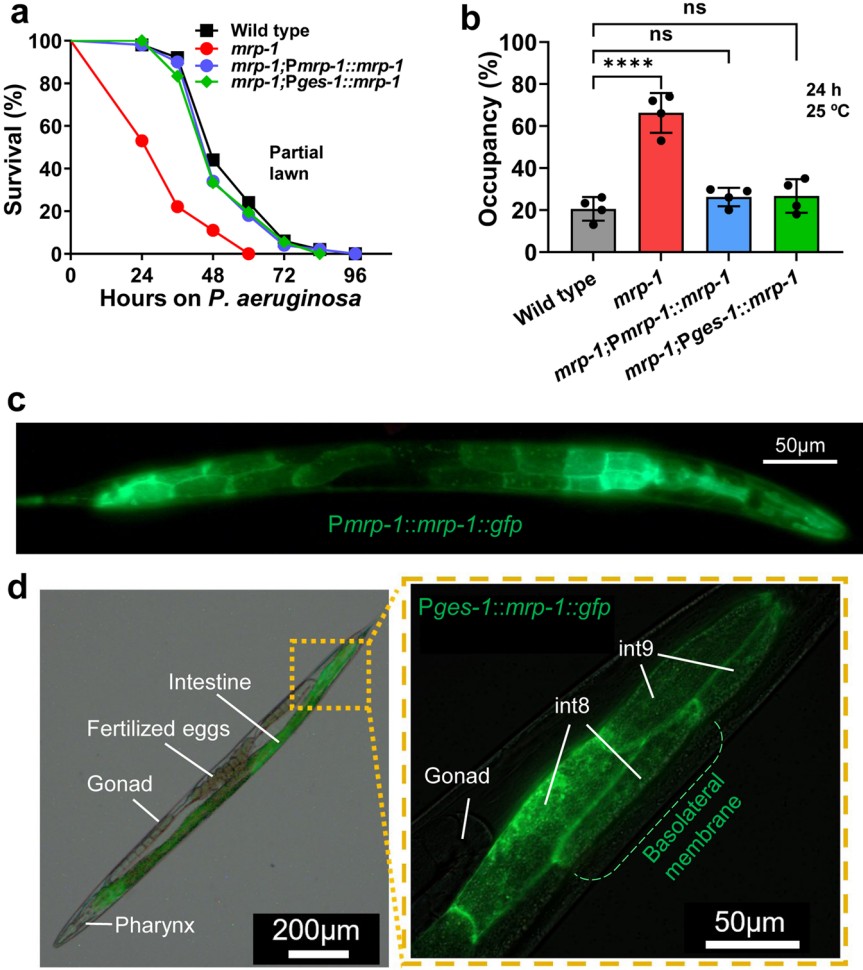

**Fig. 2 MRP-1 functions at the basolateral membrane of *C. elegans* intestinal cells. a** Representative survival plot of *C. elegans* strains exposed to a partial lawn of *P. aeruginosa*. The survival rates of *mrp-1;Pmrp-1::mrp-1* (ns) and *mrp-1; Pges-1::mrp-1* transgenic animals were compared to that of wild type using the log-rank (Mantel–Cox) test, and the differences were found nonsignificant, ns, *P* = 0.3965 and ns, *P* = 0.3205, respectively. **b** Lawn occupancy of *C. elegans* strains on a partial lawn of *P. aeruginosa* at 24 h. The bars represent the mean ± SD from four independent experiments. Assays were performed at 25 °C, with 40–50 animals/assay. ****$P \leq 0.0001$ via one-way ANOVA with Dunnett's test. 'ns' indicates nonsignificant; '*' indicates a significant difference. **c**, **d** Representative photomicrographs of MRP-1::GFP transgenic animals. Rescue of the *mrp-1* gene mutation with translationally fused MRP-1::GFP expressed under the control of (**c**) P*mrp-1* or (**d**) P*ges-1*, the intestine-specific promoter. Scale bars are provided.

indicate that *mrp-1* is required for a proper host response to *P. aeruginosa* infection.

**MRP-1 functions in the basolateral membrane of intestinal cells to protect *C. elegans* from *P. aeruginosa* infection.** The *mrp-1* gene is expressed in several structures, including the intestine, pharyngeal–intestinal valve, pharynx, nervous tissue, rectal valve, and vulva[6]. We were interested in identifying the foci of activity of the *mrp-1* gene during pathogen infection. Thus, we generated a transgenic *mrp-1(pk89)* strain in which the MRP-1 protein was translationally fused to green fluorescence protein (GFP) (*mrp-1;Pmrp-1::mrp-1::gfp*). We also generated a tissue-specific rescue strain expressing MRP-1::GFP under the control of the intestinal-specific *ges-1* promoter (*mrp-1;Pges-1::mrp-1::gfp*) (Supplementary Fig. 4). The rescue of *mrp-1* driven by its own promoter or the intestine-specific promoter fully restored the phenotype to a wild-type level of resistance against *P. aeruginosa* (Fig. 2a). Furthermore, the fractions of rescued worms occupying the pathogen lawn were comparable to those of the wild type (Fig. 2b).

The pseudocoelomic cavity serves as a circulatory system for *C. elegans*, and the signaling among several cell types is mediated

through this cavity[41,42]. It has been suggested that the internal state of the animals can be sampled by neurons, such as AQR, PQR, and URX, which have sensory cilia that directly contact the fluid of the pseudocoelomic cavity[29,43]. As shown in Fig. 2c, the MRP-1 protein localized at the cell surface and was highly concentrated in the basal and lateral regions of intestinal cells (Fig. 2d). To further explore the expression of MRP-1, we employed a fusion protein marker, VHA-6::dsRED, which sublocalizes at the apical region of the intestinal cells[44]. We generated a transgenic animal (N2 wild type;P*vha-6::vha-6::dsred* + P*ges-1::mrp-1::gfp*) that has the apical and basolateral region of the intestinal cells labeled red (VHA-6::dsRED) and green (MRP-1::GFP), respectively (Supplementary Fig. 5). We observed that the two fluorescence protein markers did not colocalize within the intestinal cells, confirming that MRP-1 is dominantly expressed at the basolateral membrane of the intestinal cells (Supplementary Fig. 5). This localization of MRP-1 at the basolateral rather than the apical membrane of the cell is conducive to the export of signals into the pseudocoelomic cavity that may alert other tissues of the presence of an intestinal infection.

It has been reported that MRP-1 also colocalizes on the outer membrane of mitochondria[45]. To investigate the potential role of mitochondria in the phenotype of *mrp-1* animals and to confirm the colocalization of MRP-1 in the outer membrane of mitochondria in *C. elegans*, we counterstained the mitochondria of nematodes with a mitochondrion-specific red fluorophore dye and determined whether fluorescence signals from the dye and MRP-1::GFP proteins overlapped. No colocalization of GFP and the red fluorescent marker was observed in the animals (Supplementary Fig. 6). Taken together, these results suggest that MRP-1 functions in the basolateral membranes of intestinal cells to control *C. elegans* defense against pathogen infection.

**The level of GSSG increases during *P. aeruginosa* infection**. We hypothesized that the avoidance defect observed in *mrp-1(pk89)* animals was due to a failure to export specific molecules released via the MRP-1 efflux pump during infection. Reduced GSH and oxidized GSSG are well-known substrates of MRP-1 and are transported or undergo co-efflux in various physiological processes and pharmacological conditions[3]. We therefore investigated which species of glutathione, GSH or GSSG, increases in abundance during pathogen infection by employing an in vivo glutathione sensor *C. elegans* strain, JV2 (jrIs2 [*rpl-17*p::Grx1-roGFP2 + *unc-119*(+)])[46]. Genetically encoded reduction–oxidation-sensitive probes such as the redox-sensitive GFP (roGFP) and its variant roGFP2 allow real-time visualization of the oxidation state of the sensor. The roGFPs have two fluorescence-excitation maxima at about 400 and 490 nm, and display rapid and reversible ratiometric changes in fluorescence in response to changes in ambient redox potential[47]. The fusion of glutaredoxin (Grx) protein with roGFP2 further increases the specificity for glutathione, enhances the kinetics of equilibration between the glutathione and roGFP2-Grx redox couples, and effectively allows measurement of the glutathione redox potential[48]. The transgenic sensor strain JV2 was generated by expressing Grx1-roGFP2 under the large ribosomal subunit 17 (*rpl-17*) gene promoter[46]. Thus, the sensor protein is expressed globally and constitutively within the cytosol of tissues of the animal.

To determine whether the roGFP sensors would respond to changes in the systemic glutathione levels, we soaked the animals in a cell-permeable GSSG methyl ester to mimic GSSG accumulation inside the body of the animals. By measuring the signal intensity of the fluorescence emitted by the Grx1-roGFP2 sensor, we quantified the relative levels of GSSG and GSH in the treated and untreated animals (Supplementary Fig. 7a, b). This process has helped us to validate and standardize our approach for detecting the changes in the redox balance of infected animals. We measured the signal intensity of the sensor in JV2 animals that were grown on nonpathogenic *E. coli* and compared it with the signal intensity in animals that were infected with *P. aeruginosa*. We found that *P. aeruginosa* infection depleted the GSH levels, while it increased the GSSG levels, resulting in a redox imbalance (Fig. 3a–d). To further examine the role of the MRP-1 efflux pump in regulating the cellular concentrations of GSSG, we measured GSSG levels in the JV2 sensor strain crossed with the *mrp-1(pk89)* mutant. We observed that the absence of the MRP-1 efflux pump also resulted in the intracellular retention of GSSG molecules during pathogen infection (Fig. 3e).

Because *P. aeruginosa* infection elicits *C. elegans* avoidance by causing intestinal distension[30–32], we investigated whether distention affects GSSG levels. The effects of intestinal distension caused by *P. aeruginosa* infection can be mimicked by disrupting the defecation motor program (DMP) of animals via RNAi targeting of genes that control defecation[32,49]. The genes *aex-5*, *flr-1*, *nhx-2*, and *pbo-1* were selected for this study as their

inactivation is known to lead to defects in the DMP, which results in the bloating and distention of the intestinal lumen by the accumulation of bacteria[50–52]. We tested whether the RNAi-mediated disruption of the DMP in animals would lead to increases in GSSG production and a redox imbalance, and we observed no increase in GSSG disruption of DMP in the JV2 roGFP-sensor strain (Supplementary Fig. 8). We also studied whether GSSG may induce *daf-7* in the ASJ chemosensory neuron pair, as *daf-7* expression increases by exposure to phenazine-1-carboxamide and pyochelin produced by *P. aeruginosa*[40]. We observed that exposure to GSSG did not increase the expression of *daf-7* in the ASJ chemosensory neuron pair (Supplementary Fig. 9).

Since the above results showed that GSSG is elevated intracellularly in response to *P. aeruginosa* infection, and since animal-behavior modulation requires nervous-system involvement, we hypothesized that in the absence of the MRP-1 efflux pump, cytosolic GSSG might fail to reach its intercellular cell targets and hence be incapable of modulating animal behavior. To test this hypothesis, we first studied the effects of GSSG supplementation on animal behavior. We performed a modified lawn-occupancy assay in which a partial lawn of *P. aeruginosa* was supplemented with GSSG or buffer as a control. We observed that the addition of GSSG to the lawn induced enhanced pathogen avoidance in the presence of live but not killed *P. aeruginosa* (Fig. 4a and Supplementary Fig. 10). Supplementation of GSSG was also capable of inducing pathogen avoidance in *mrp-1(pk89)* mutant animals (Fig. 4b).

We also studied the preference of infected animals for GSH or GSSG. On modified choice-assay plates containing small lawns of *P. aeruginosa* or *E. coli*, droplets of solution containing equimolar concentrations of GSH or GSSG were added to the top of the bacterial lawns (Fig. 4c). The animals were then placed at the center of the assay plate at a position equidistant from the two bacterial lawns and allowed to move freely for 3 h. We observed that on a choice-assay plate containing *P. aeruginosa*, the preference of the animals shifted toward the lawn supplemented with GSH, while animals showed aversive behavior toward the lawn of *P. aeruginosa* supplemented with GSSG (Fig. 4d). This finding suggested that the lawn-leaving behavior observed in the previous exogenous GSSG-supplementation assay was not due to a physical effect such as a high- salt concentration, but was instead due to a specific effect of the GSSG molecule on the host–pathogen interaction.

One unique property of alarmins is that they are context dependent[53]. They do not exist in a pro-active form, and the locations and concentrations of these molecules are important factors in their key activation functions[54]. We next explored the possibility that the GSSG molecule functions as an alarmin that requires the presence of pathogenic bacteria to promote lawn-leaving behavior in animals. We determined the occupancy of animals on lawns of nonpathogenic *E. coli* supplemented with GSSG. In contrast to the GSSG-enhanced *P. aeruginosa* avoidance observed (Fig. 4a), we observed no GSSG-mediated lawn-leaving behavior upon interaction with nonpathogenic *E. coli* (Fig. 4e–g). Even higher GSSG concentrations than those capable of inducing avoidance of *P. aeruginosa* also failed to induce avoidance of *E. coli* (Supplementary Fig. 11), suggesting that the ability of GSSG to induce aversive behavior was pathogen dependent. These findings further suggested that the presence of the GSSG molecule alone was not the driving force of the GSSG-mediated avoidance behavior. Taken together, these results showed that GSSG is a candidate alarmin molecule produced in response to certain infections and requires a live pathogenic interaction to modulate the behavioral immunity of an animal.

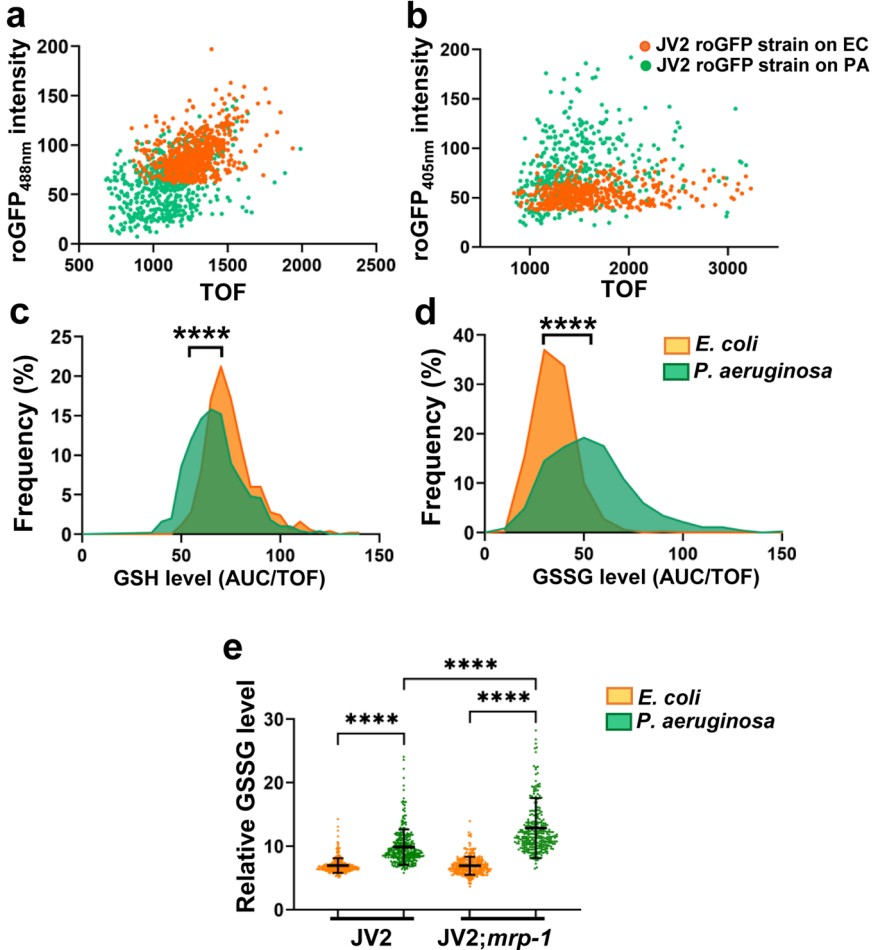

**Fig. 3 *P. aeruginosa* infection induces a redox imbalance and drives equilibrium toward increased GSSG production. a, b** Dot-plot representation of the levels of glutathione species in the sensor strain of *C. elegans* (JV2, jrIs2 [*rpl-17*p::Grx1-roGFP2 + *unc-119*(+)]) after 4–5 h of exposure to *P. aeruginosa* (PA) or *E. coli* (EC) at 25 °C, showing the roGFP2 fluorescence intensity versus time of flight (TOF). Excitation of the roGFP sensor at a wavelength of 488 nm or 405 nm allowed the quantification of GSH or GSSG, respectively. **c, d** Frequency distribution versus GSH or GSSG levels (area under the curve (AUC) x 1000 over time of flight (TOF)) in animals exposed to *P. aeruginosa* or *E. coli*. ****$P \leq 0.0001$ according to a two-tailed t-test, with 400–500 animals/assay. **e** Relative GSSG levels in the JV2 roGFP-sensor strain and JV2;*mrp-1*(pk89) animals after 4 h of exposure to *P. aeruginosa* or *E. coli* at 25 °C. Bars represent the mean ± SD, ****$P \leq 0.0001$ according to one-way ANOVA with Tukey's multiple-comparison test, with 250–300 animals/assay; '*' indicates a significant difference.

**The NMR-1 receptor is required for GSSG-mediated adult-learned aversion.** It has been demonstrated that exposure to a pathogen can train naive adult *C. elegans* to avoid the pathogen during a second encounter[55–57]. For example, prior exposure to pathogenic *P. aeruginosa* reverses the preference of *C. elegans* that is otherwise inherently attracted to a lawn of *P. aeruginosa*, and training or pre-exposure of animals can switch this preference to nonpathogenic *E. coli*[56]. To determine whether exposure to GSSG could also train naive animals to avoid pathogens, we soaked wild-type animals in a solution of cell-permeable GSSG for 3 h at 20 °C and tested their preference for *E. coli* versus *P. aeruginosa* (Fig. 5a). In a binary-choice assay, a choice index of −1.0 represented an absolute preference for *E. coli* (control bacterium), an index of 1.0 represented an absolute preference for *P. aeruginosa*, and an index of zero (0) represented an equal preference for the two food sources. We observed that exposure to GSSG shifted the preference toward *E. coli*, while naive animals that were exposed to the buffer-only control did not exhibit the aversive phenotype (Fig. 5b, c). These findings suggest that GSSG may act as a neuromodulator that is capable of warning naive animals to anticipate unfavorable future events, such as

encounters with a pathogen, and inducing aversive learning toward the pathogen.

Notably, both GSH and GSSG can stimulate N-methyl-D-aspartate (NMDA)-mediated calcium entry into dissociated neurons from rats[58]. Additionally, GSH or GSSG are potential candidate ligands of NMDA receptors (NMDARs) because the constituent amino acid sequence bears a close structural resemblance to those of NMDAR activators[59]. Glycine, for example, is a co-agonist in the activation of NMDARs by glutamate[59]. In *C. elegans*, there are 2 NMDA subunits, NMR-1 and NMR-2, that are required for the function of neural circuits and behavior[60], and it has been reported that NMR-1 is important in the memory and learning of chemotaxis, as well as adult-learned pathogen aversion[57,61]. Therefore, to test the hypothesis that NMR-1 is involved in GSSG-mediated pathogen aversion, we exposed naive *nmr-1(ak4)* mutant animals to M9 control buffer or GSSG solution and assessed their preference between lawns of pathogenic and nonpathogenic bacteria. We observed that *nmr-1(ak4)* animals did not acquire GSSG-mediated learned pathogen avoidance (Fig. 5d, e). Notably, we observed that *nmr-1(ak4)* animals displayed a wild-type level of preference toward *P. aeruginosa* (Fig. 5d), indicating that the

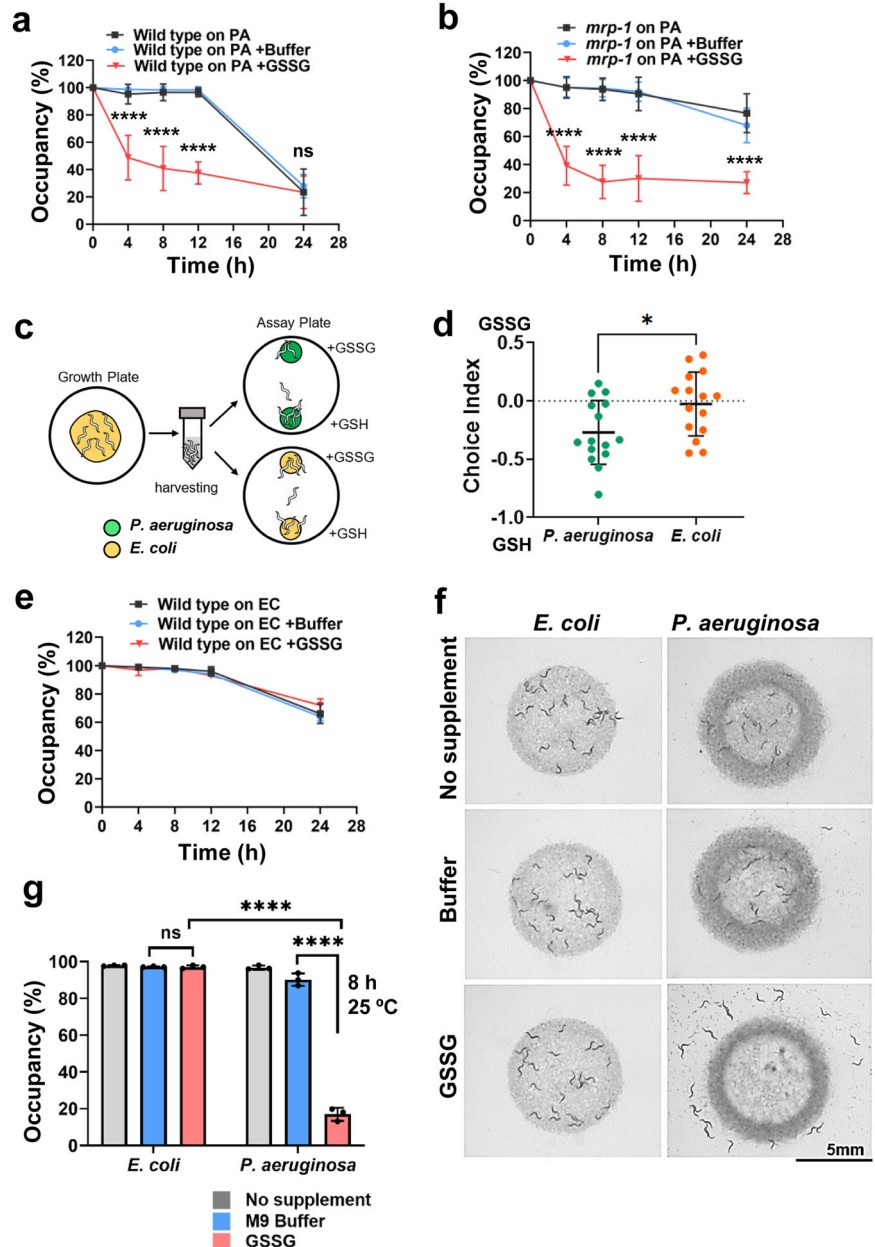

**Fig. 4 Exogenous supplementation with glutathione species modulates animal behavior via pathogenic interactions. a** Time course of the lawn occupancy of wild-type animals on a partial *P. aeruginosa* (PA) lawn with or without GSSG supplementation. **b** Time course of the lawn occupancy of *mrp-1(pk89)* animals on a partial PA lawn with or without GSSG supplementation. 'ns' indicates nonsignificant; '*' indicates a significant difference; ****$P \leq 0.0001$ according to two-way ANOVA from three independent experiments, with 40–50 animals/assay; error bar represents the standard deviation (**a**, **b**). **c** Schematic representation of the glutathione-supplementation- modified two-choice assay. Synchronized animals were harvested and transferred to choice-assay plates containing a small lawn of *P. aeruginosa* or *E. coli*. Solutions of equimolar concentrations of GSH and GSSG were added on top of the bacterial lawns. The animals were placed at the center of the plate equidistant from the lawns. The number of animals on each bacterial lawn was counted 3 h after transfer. **d** Choice preference of wild-type animals on *P. aeruginosa* or *E. coli* supplemented with GSH and GSSG. *$P \leq 0.05$ via a two-tailed *t*-test, with 50–150 animals/assay; each dot represents a replicate from five independent experiments. **e** Representative time course of the lawn occupancy of wild-type animals on partial lawns of *E. coli* (EC) supplemented with M9 buffer, GSSG dissolved in M9 buffer, or the unsupplemented control. **f** Representative images of the lawn occupancy of wild-type animals on partial lawns of *E. coli* or *P. aeruginosa* with varying lawn conditions. Images were acquired at 8 h. **g** Lawn occupancy of wild-type animals on a partial lawn of *E. coli* or *P. aeruginosa* at 8 h. Each dot represents the average occupancy determined from three independent assays performed at 25 °C with 40–50 animals/assay. 'ns' indicates nonsignificant; '*' indicates a significant difference; ****$P \leq 0.0001$ according to one-way ANOVA followed by Tukey's multiple-comparison test. Error bar represents the standard deviation (**d**, **e**, **g**).

learning defect of *nmr-1(ak4)* mutant animals was not due to defective pathogen sensing. However, the lawn occupancy of *nmr-1(ak4)* mutants was not different than that of wild-type animals (Supplementary Fig. 12a). These results suggest that NMR-1 is not the receptor that is involved in the lawn occupancy of naive animals, however, it is a receptor required for learned pathogen aversion as well as GSSG-mediated adult-learned pathogen aversion.

To identify the sets of neurons involved in GSSG–NMR-1-mediated signaling, we constructed transgenic strains expressing

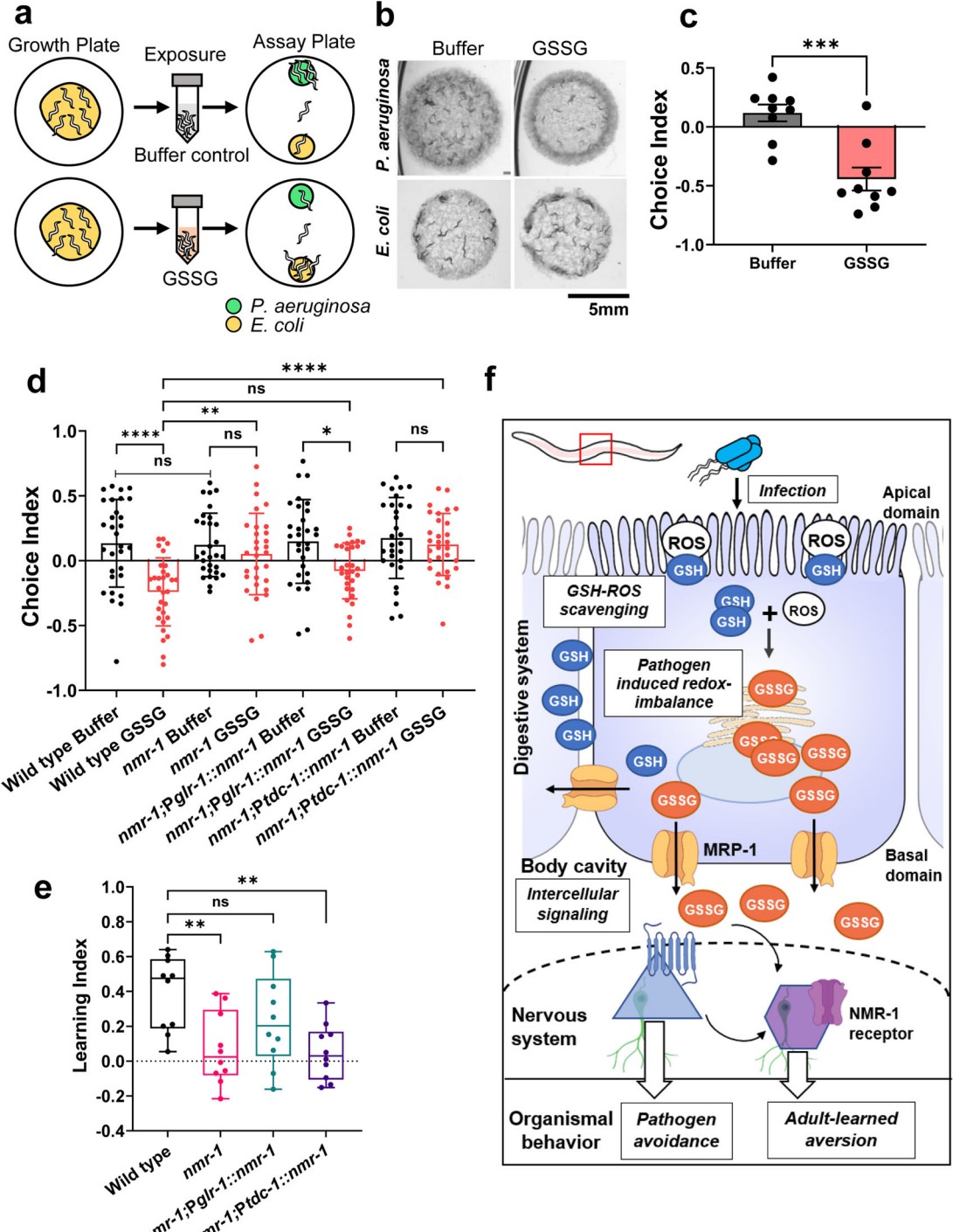

the NMR-1 receptor under heterologous *glr-1* and *tdc-1* gene promoters (Supplementary Fig. 12b). We observed that the GSSG-mediated learning defect in the *nmr-1(ak4)* mutant was rescued by the expression of *nmr-1* driven by the *glr-1* promoter but not the *tdc-1* promoter (Fig. 5d, e). We also observed similar choice-preference and learning profiles in these animals in the context of pathogen-mediated training (Supplementary Fig. 12c, d). The NMR-1 receptor is expressed in several interneurons, gonadal sheath cells, and oocytes. The expression of NMR-1 using the heterologous *glr-1* promoter helped us to exclude the possible involvement of the germline in this GSSG-mediated learned aversion because the *glr-1* gene is expressed in most NMR-1-expressing interneurons, such AVA, AVD, and AVE neurons[61],

though not in the germline. Furthermore, based on the expression specificity of the *nmr-1*, *glr-1*, and *tdc-1* promoters in neurons, we also determined that at least one of the interneurons, AVD, AVG, PVC, or their combinations, was required for GSSG-mediated learned aversion. However, it remains unclear at this point whether endogenous GSSG molecules drove the learned aversion behavior directly through an interaction with the NMR-1 receptor or indirectly through other neurons via neural circuit networks. It has been reported that NMR-1-expressing interneurons, such as PVC interneurons, may sense stimuli directly or indirectly through other neurons[62]. Taken together, these findings revealed that the NMR-1 receptor is important for GSSG-mediated learned pathogen aversion.

**Fig. 5 Prior exposure to GSSG induces training of naive animals to avoid pathogens. a** Schematic representation of the GSSG-mediated priming of naive wild-type animals and two-choice preference assays. Synchronized adult animals were harvested from growth plates and soaked in M9 buffer or M9 buffer containing GSSG for 3 h with regular tapping agitation. Soaked animals were then transferred to assay plates containing small lawns of *P. aeruginosa* or *E. coli*. The number of animals on each lawn was then counted after 3 h. The assay was performed at 20 °C, using 50–150 animals/assay. **b** Representative images depicting the differential preferences of animals exposed to GSSG and those exposed to the buffer control. **c** Choice-preference index of wild-type animals exposed to GSSG or the M9 control buffer. A positive index indicates attraction, while a negative index indicates aversion to *P. aeruginosa*. Each dot represents a biological replicate from three independent experiments, ***$P \leq 0.0005$ according to the two-tailed t-test, with 100–200 animals/assay. **d** Choice-preference index of *C. elegans* strains exposed to GSSG or the M9 control buffer. Synchronized 1-day-old animals were soaked in GSSG solution or buffer alone for 3 h at 20 °C. Each dot represents a biological replicate from 10 independent experiments, with 100–200 animals/assay. Bars represent the mean ± SD (**c**, **d**). **e** Learning index after adult exposure to GSSG. Boxes represent the median and first and third quartiles, and whiskers represent the 10th–90th percentiles. *P*-values were generated by one-way ANOVA with Tukey's multiple-comparison test (**d**) or by one-way ANOVA with Dunnett's correction (**e**). 'ns' indicates nonsignificant; '*' indicates a significant difference; *$P \leq 0.05$, **$P \leq 0.005$, and ****$P \leq 0.0001$. **f** Model for pathogen-driven redox imbalance and export of the GSSG danger signal to modulate host behavioral immunity. During pathogen infection, increased generation of ROS results in GSH depletion and excess GSSG formation, leading to cellular-redox imbalance. MRP-1 has a higher affinity for GSSG than GSH and can readily transport GSSG into the extracellular milieu. This extracellular GSSG may interact with target receptors and tissues via a cell nonautonomous action to control host immune responses.

## Discussion

The export of danger signals through broad-specificity efflux pumps during cellular insults may constitute an early warning system because this process does not rely on extensive cell membrane damage to release danger signals. The present study demonstrates a potential anatomical basis for and molecular mechanism of the regulation of the flight response in *C. elegans* nematodes during infection by the efflux MRP-1 protein (Fig. 5f). We observed that the MRP-1 protein is located in the basolateral domain of intestinal cells, from which it may mediate the export of GSSG that travels through the pseudocoelomic cavity to alert other tissues to the presence of an infection. Notably, there are several other ABC transporters present in *C. elegans*. ABC transporters, such as P-glycoproteins (PGPs), are associated with sensitivity to environmental insults, pharmacoactive drugs, and pathogens[63]. It is not known whether PGP efflux pumps may also participate in the export of danger signals. However, it is highly unlikely that the molecules exported via PGP pumps would reach the nervous system because the pumps are expressed on the apical domain of the intestine and will effectively transport molecules into the intestinal lumen[64].

Several lines of evidence indicate that redox signaling is one of the many regulators of *C. elegans* behavior in response to infectious pathogens[65,66]. *C. elegans*, in response to microbial infection, produces several reactive oxygen species (ROS)[67,68]. The dual oxidase Ce-Duox1/BLI-3, a member of the NADPH-oxidase family, is responsible for ROS generation in *C. elegans*[67,69]. It has been reported that elevated ROS may result in oxidative stress, leading to cellular injury and damage. However, to mount appropriate counteractive and restorative responses, the cell requires a functioning redox signaling, which is regulated by the different ROS species[70]. By-products of redox imbalance, such as accumulation of GSSG molecules, in combination with pathogens, may serve as alarmins that alert the animal of an impending danger.

The ability of an organism to anticipate impending danger based on a previous encounter confers a great survival benefit. Even simple life forms such as nematodes learn from previous exposure to a chemical or pathogen and tend to avoid exposure to these agents in subsequent encounters, indicating an ability to form a relevant memory, and then recall it under certain circumstances. *C. elegans* has evolved a mechanism to learn from experience and avoid pathogens[56]. It has been reported that the *C. elegans* NMDA-type receptor NMR-1 is important for adult-learned aversion[57]. NMR-1 belongs to the glutamate-receptor family, which is important in memory, plasticity, and learning[61]. In the present study, we also demonstrated that the exposure of

naive worms to GSSG could induce aversive learning. GSSG likely interacts with the NMR-1 receptor via allosteric regulation or redox modulation, however, it is unclear at this point whether such binding results in agonistic or antagonistic activity in the set of NMR-1-expressing neurons.

The implications of our findings may not be limited to the behavioral immunity of *C. elegans* because it is highly likely that similar effects may be observed in higher animals based on the presence of analogous peptides and receptors constituting glutathione–NMR-1 signaling pathways. It has been reported that the chronic depletion of GSH in *Gclm*(−/−) knockout mice results in decreased anxiety and reduced fear-associated behaviors compared with those in wild-type mice[71]. Additionally, the mechanism of action of ketamine as an antidepressant involves synaptic or extrasynaptic NMDAR inhibition[72]. Furthermore, abnormalities in glutamatergic neurotransmission or glutamatergic dysfunction play an important role in the development of many major psychiatric conditions[73]. Recent evidence suggests that cultured glial cells may release glutathione into the surrounding medium and probably in vivo within the brain[74]. Hence, taking into account this indirect evidence, it is tempting to consider an important role of the GSSG–NMR-1 signaling pathway in normal and abnormal brain activity, especially during redox imbalances generated in the presence of inflammation or infections.

In conclusion, our results demonstrated that the gut efflux pump MRP-1 functions as an important component of resistance against pathogen infection in the nematode *C. elegans*. We identified the MRP-1-specific substrate endogenous GSSG as the signaling molecule involved in modulating the behavior of the animals. Pathogen infections, or oxidative stressors that can lead to redox imbalance, may universally affect this signaling pathway. Our study also suggested that the GSSG molecule could serve as a signaling agent, as exposure to the molecule resulted in an induction of aversive learning via the NMR-1 receptor. Notably, MRP-1 of *C. elegans* shows a high degree of sequence homology to human MRP-1, with a percent identity of 45.79% (protein–protein BLAST[75]), and there is also evidence that human MRP-1 can rescue *mrp-1*-associated phenotypes in a *C. elegans* mutant strain[76]. Thus, the mechanism of endogenous inflammatory by-product transport described here may be a general mechanism used by metazoans to sense the presence of a pathogen and activate immune defenses.

## Methods

**Material availability**. All unique reagents generated in this study are available from the corresponding author under a completed Materials Transfer Agreement.

**Nematode and bacterial strains**. *C. elegans* strains were cultured and maintained under standard conditions on *E. coli* OP50 at 20 °C, unless otherwise indicated. The Bristol N2 strain was used as the wild-type control, unless otherwise indicated. The following strains were also used: NL147 *mrp-1(pk89)*, RB1713 *mrp-2(ok2157)*, RB1028 *mrp-3(ok955)*, VC712 *mrp-4(ok1095)*, RB1070 *mrp-6(ok1027)*, RB1269 *mrp-8(ok1360)*, JV2 jrIs2 [*rpl-17p*::Grx1-roGFP2 + *unc-119*(+)], FK181 ksIs2 [*daf-7p*::GFP + *rol-6(su1006)*], and VM487 *nmr-1(ak4)*. The *C. elegans* strains were obtained from the *Caenorhabditis* Genetics Center (University of Minnesota, Minneapolis, MN). The following bacterial strains were used: *E. coli* OP50, *E. coli* HT115(DE3), *E. coli* TOP10, *P. aeruginosa* PA14[36], and *P. aeruginosa* PA14 Δphz. Bacterial cultures were grown on Luria-Bertani (LB) broth or LB plates at 37 °C.

**RNA interference (RNAi)**. RNAi was used to generate loss-of-function RNAi phenotypes by feeding nematodes *E. coli* expressing double-stranded RNA (dsRNA) homologous to a target gene[77]. RNAi was carried out as described previously[31]. Briefly, *E. coli* cells harboring appropriate vectors were grown in LB broth containing ampicillin (100 mg/ml) and tetracycline (12.5 mg/ml) at 37 °C overnight and plated on nematode growth medium (NGM) plates containing 100 mg/ml ampicillin and 3 mM isopropyl-b-D-thiogalactoside (IPTG). RNAi-expressing bacteria were allowed to grow overnight at 20 °C. Gravid adults were transferred to RNAi-expressing bacterial lawns and allowed to lay eggs for 2 h. Gravid adults were removed, and their eggs were allowed to develop at 20 °C, until reaching a particular developmental stage required for subsequent assays. For the knockdown of genes that cause embryonic lethality, RNAi was started at L4-stage animals, and the animals were allowed to develop further for 48 h at 20 °C or otherwise indicated. Control animals were grown on empty vector-containing bacteria in all cases. All RNAi clones were verified by DNA sequencing.

***C. elegans* killing assay**. The *C. elegans* killing assay was performed according to a previously described protocol[31], with minor modifications. Bacterial cultures were grown by inoculating individual bacterial colonies into 2 ml of LB broth and incubating them for 16 h on a shaker at 37 °C. For partial lawn-mediated killing assays, 10 μl of a bacterial culture was dispensed in the center of a 3.5-cm-diameter standard slow-killing plate (SK plate, modified NGM agar (0.35% instead of 0.25% peptone)); to reduce variations due to excess moisture in the assays, the SK plates were held at room temperature for at least 72 h after pouring and preparation. The assay plates were allowed to air-dry for 1–2 h and were then incubated at 37 °C for 16 h. For full lawn-mediated killing assays, 20 μl of a bacterial culture was spread evenly on the surface of a 3.5-cm-diameter SK plate. The plates were allowed to air-dry for 1–2 h under laminar flow cabinet and were then incubated at 37 °C for 16 h. Before the transfer of synchronized gravid adult animals, the assay plates were cooled at room temperature for 1 h. For partial lawn-mediated killing assays, the animals were placed directly at the center of the bacterial lawn. For the subsequent daily transfer, the fraction of animals that were found "inside or outside" of the bacterial lawn were placed accordingly to the corresponding area of the assay plate. This approach improves the accuracy of the assay as it ensures a survival advantage to the fraction of animals that are effectively avoiding the pathogen. The killing assays were performed at 25 °C, and live animals were transferred daily to fresh killing plates. The animals were scored at the indicated times and were considered to be dead when they failed to respond to touch.

**Pathogen-avoidance assays**. Pathogen-avoidance assays were performed according to a previously described method[32], with minor modifications. The bacterial cultures were grown by inoculating individual bacterial colonies into 2 ml of LB broth and incubating them for 16 h on a shaker at 37 °C. Then, 10 μl of the culture was plated in the center of a 3.5-cm-diameter SK plate (prepared as described above) and allowed to air-dry for at least 1–2 h under laminar flow cabinet. The assay plates were incubated at 37 °C for 16 h and cooled at room temperature for 1 h before they were used for the assays. A total of 40–50 synchronized gravid adult hermaphroditic animals were split into groups of 20–25 animals per test plate and were directly placed on the lawns of test bacteria.

For the glutathione-supplementation avoidance assay, the assay plates were prepared as mentioned above. In addition, 10 μl of 200 mM GSSG (glutathione, oxidized, free acid; Sigma-Aldrich, 3542-1GM) reconstituted in M9 buffer was added directly on top of the partial bacterial lawns to cover their entire surface. To avoid extreme pH changes in the supplement solution, GSSG should be reconstituted in an appropriate buffer system. It is also critical to cover the surface of the bacterial lawn as much as possible with the supplement solutions as the animals tend to stay on the uncovered portion of the lawn. The plates were allowed to air-dry for 1 h. Synchronized gravid adult animals were directly placed on the lawn and were observed for lawn-avoidance behavior. The numbers of animals on and off the lawns were counted at the specified times for each experiment. The experiments were performed at 25 °C. The percent occupancy was calculated using the following equation:

$$\text{Occupancy} (\%) = \left[\frac{\text{number of animals on lawn}}{\text{total number of animals}}\right] \times 100 \quad (1)$$

**Generation of transgenic animals**. To rescue the *mrp-1(pk89)* mutation, the pJL1 (pPD95.77_P*mrp-1*::*mrp-1*::SL2::*gfp*) plasmid was constructed by cloning the 3-kb promoter region of the *mrp-1* gene fused to the cDNA of *mrp-1* isoform C using the *SphI-XmaI* restriction sites. pJL1 was constructed in the pPD95.77 vector backbone (Fire Lab *C. elegans* Vector Kit; Addgene, Cambridge, MA), wherein the translational fusion of MRP-1 and GFP was avoided by the inclusion of the trans-splicing sequence SL2[39]. For GFP labeling and the generation of translational fusion constructs, the *mrp-1*::*gfp* cassette was expressed using 3 kb of the promoter region of *mrp-1* or 3.3 kb of the promoter region of the intestine-specific gene *ges-1* to generate the pJL2 (pPD95.75_P*mrp-1*::*mrp-1*::*gfp*) and pJL3 (pPD95.75_P*ges-1*::*mrp-1*::*gfp*) plasmids, respectively. *mrp-1* cDNA was prepared by reverse transcription using oligo dT primers (RevertAid First Strand cDNA Synthesis Kit; Thermo Scientific K1621). The *mrp-1* gene fragment was inserted into the pPD95.75 backbone vector using the *SphI-KpnI* restriction sites. The primers used for the construction of the above plasmids are (uppercase letters represent restriction endonuclease (RE) sequence): pJL1, Pmrp-1 SphI F 5′GCATGCccagtcaacgtttcgataagttg, Pmrp-1 ApaI R 5′ GGGGCCCttcttaattggctcggttcggt, mrp-1 CDS ApaI F 5′GGGGCCCCatgttcccgttagttcgt-gaac, mrp-1 CDS XmaI R 5′CCCGGGCtacacaacatttgcatctttttgc. pJL2, Pmrp-1 SphI fusion F 5′actgGCATGCccagtcaacgtttcgataagttgg, mrp-1 KpnI fusion R 5′actgGG-TACCcccacaacatttgcatctttgccattgaatag. pJL3, Pges-1 SphI F 5′acgtGCATGCttattttca-gatcagtaattcgaaatgtttctactggaat, Pges-1 ApaI R 5′ actcGGGGCCCctgaattcaaagatagatatgtaatagattttgaagcc.

For studying MRP-1 subcellular localization, pJL6 (coel::RFP_P*vha-6*::*vha-6*::*dsred*) was constructed by inserting the coding sequence of *vha-6* that includes a 894-bp upstream-promoter region of the *vha-6* gene into a coel::RFP (Piali Sengupta lab plasmids; Addgene, Cambridge, MA) backbone vector via *SphI-AgeI* restriction sites. The stop codon of *vha-6* gene was replaced with a GS linker, and the 6th exon segment was fused to the open-reading frame of dsRED of the backbone vector. The plasmid constructs, pJL3 (pPD95.75_P*ges-1*::*mrp-1*::*gfp*), pJL6 (coel::RFP_P*vha-6*::*vha-6*::*dsred*), and coel::RFP con-injection marker were then micro-injected to N2 strain. The primers used for the construction of the above plasmids are (uppercase letters represent RE sequence): pJL6, Pvha-6 SphI F 5′actGCATGCagcacagaactgcattaagtatac, vha-6 AgeI R 5′ actACCGGTCtaccagaaccaccgtgatcgagtccctcgtaaacg.

To rescue the *nmr-1(ak4)* mutation, the pJL4 (pPD95.77_P*glr-1*::*nmr-1*::SL2::*gfp*) and pJL5 (pPD95.77_P*tdc-1*::*nmr-1*::SL2::*gfp*) plasmids were constructed by cloning the 4.1-kb upstream sequences of the *glr-1* and *tdc-1* genes, respectively. The heterologous promoters were fused to the cDNA of the *nmr-1* gene using the *SphI-PstI* restriction sites. The gene fragments were inserted into the pPD95.77_SL2 backbone vector using the *SphI-BamHI* restriction sites. All constructed plasmids were subjected to DNA sequencing verification. Transgenic strains were developed by the microinjection of 40 ng/μl plasmid together with 50 ng/μl coinjection marker (coel::RFP plasmid) and 10 ng/μl carrier plasmid DNA (pUC18). The primers used for the construction of the above plasmids are (uppercase letters represent RE sequence): pJL4, Pglr-1 SphI F 5′tacGCATGCcaataatctcagttttttcagccaatcaca, Pglr-1 PstI R 5′tacCTGCAGtgtgaatgtgtcagattgggtgcc, NMR-1 PstI CE Kozak F 5′actCTGCAG ggtagaaaaaaatgttccgaatatcagttatatttatttggatatttttac, NMR-1 BamHI R 5′actGGATCC gttacacataaaatctagttgatcttgctcttcgaaacc. pJL5, Ptdc-1 SphI F 5′tacGCATGCagtaaccgtt cgaactatttccatt, Ptdc-1 PstI R 5′tacCTGCAGttgggcggtcctgaaaaatgca.

**Determination of GSH–GSSG profiles in animals**. The roGFP-sensor strain JV2 jrIs2 [*rpl-17p*::Grx1-roGFP2 + *unc-119*(+)] animals were synchronized on NGM plates containing *E. coli*. For the standardization of relative GSH–GSSG quantification assay, young gravid adult animals were transferred to a snap-cap tube containing S-basal medium and 5% saturated *E. coli* OP50 broth culture, which was supplemented with 25 mM (final concentration) permeable methyl-GSSG, or a snap-cap tube containing S-basal medium without the supplementation. The animals were incubated in a volume of 100 μl of medium at 25 °C for 4–5 h, with frequent tapping agitation. The animals were washed thrice with M9 buffer, and the GSH and GSSG profiles of the animals were determined using a COPAS Biosort flow cytometer (Union Biometrica Inc.). The animals were gated based on the extinction and time-of-flight (TOF) parameters. To measure GSSG levels, a 405-nm excitation laser was used along with a 405-nm band-pass excitation filter with a neutral density (ND) of 0.6. Using this configuration, the instrument settings were optimized, and the emission data were collected via a 512/25-nm band-pass emission filter (green channel). The biosorter acquisition parameters were set as follows: extinction gains—100.0, 512/25 green gains—6.0, PMT volts—700, and 405-nm laser power—100 mW. After the measurement of the GSSG profiles, the animals were harvested again through bulk sorting for the subsequent determination of the GSH profiles. To measure GSH levels, a 488-nm excitation laser was used along with a 488-nm band-pass excitation filter with an ND of 1.5. Using this configuration, the instrument settings were optimized, and the emission data were collected using a 512/25-nm band-pass emission filter (green channel). The biosorter acquisition parameters were set as follows: extinction gains—2.0, 512/25 green gains—2.0, PMT volts 390, and 488-nm laser power—50 mW. For the measurement of GSH or GSSG profiles of animals exposed to food or pathogen, young gravid adult animals were harvested in M9 buffer and transferred to a 6-cm NGM plate containing a 16-h lawn of *P. aeruginosa* (PA14) or control *E. coli* (OP50). After 4–5 h of incubation at 25 °C, the animals were collected and washed

thrice with M9 buffer to remove bacteria and debris. The GSH or GSSG profiles of the animals were determined using a COPAS Biosort flow cytometer. The acquired data were exported into Excel files and subjected to analysis using Prism 9 (GraphPad).

**Two-choice preference assays**. A two-choice preference assay was conducted as previously described[31,56], with minor modifications. *P. aeruginosa* (PA14) and *E. coli* (OP50) cultures were grown in 3 ml of LB broth for 16 h at 37 °C with shaking. Then, 10 µl of each culture was placed at opposite positions on 6-cm-diameter NGM plates to generate small partial lawns. The plates were dried for 30 min, incubated at 37 °C for 16 h, and allowed to cool at room temperature for at least 1 h before transferring the animals. For GSSG priming and exposure, synchronized young gravid adult animals grown on *E. coli* were harvested, washed, and submerged in 100 µl of M9 buffer only (control) or M9 buffer containing 50 mM of the cell-permeable glutathione disulfide methyl ester (MyBioSource, MBS407313). To avoid extreme pH changes in the supplement solution, GSSG should be reconstituted in an appropriate buffer system. The animals were incubated in the solutions for 3 h at 20 °C with periodic tapping agitation at 30-min intervals. To remove the excess GSSG solution, the animals were centrifuged at $60 \times g$ for 30 s and rinsed twice with M9 buffer. From this point onward, the assay was performed quickly to avoid the inter-conversion of glutathione molecules by active host enzymes, which could reduce the efficacy of exposure. We transferred 30–70 animals to the center of each plate at a position equidistant from the two bacterial lawns. The animals were allowed to move freely for 3 h at 20 °C, and the numbers of animals on both lawns were counted. A choice index (CI) was calculated using the following equation:

$$CI = \frac{[(\text{No. of worms on } P. \, aeruginosa) - (\text{No. of worms on } E. \, coli)]}{[(\text{No. of worms on } P. \, aeruginosa) + (\text{No. of worms on } E. \, coli)]} \quad (2)$$

An adult-learned aversion assay was conducted as previously described[56], with minor modifications. Synchronized 1-day-old adult gravid animals were used for the assay, and the experiment was further performed as described above. The CI was calculated, and the learning index was obtained by subtracting the CI of the animals exposed to GSSG or a pathogen from the CI of naive animals.

For the GSH and GSSG molecule preference assay, the choice-assay plate was prepared as described above but with partial lawns of the same bacterium (*P. aeruginosa* or *E. coli*) at both ends. Aliquots of 10–12 microliters each of 200 mM GSH prepared with M9 buffer (glutathione, reduced, free acid; Sigma-Aldrich 3541-5GM) and GSSG (glutathione, oxidized, free acid; Sigma-Aldrich 3542-1GM) were placed directly on top of the bacterial lawns. To avoid extreme pH changes in the supplement solutions, GSH or GSSG should be reconstituted in an appropriate buffer system. The plate was allowed to air-dry inside a laminar flow cabinet for 1 h at room temperature, and naive N2 animals were transferred to the center of the plate at a position equidistant from the two bacterial lawns. The animals were allowed to move freely, and the numbers of animals in each zone were counted. The experiments were performed at 20 °C. A positive CI indicated a preference for GSSG, while a negative CI indicated a preference for GSH. A choice index (CI) was calculated using the following equation:

$$CI = \frac{[(\text{No. of worms on GSSG}) - (\text{No. of worms on GSH})]}{[(\text{No. of worms on GSSG}) + (\text{No. of worms on GSH})]} \quad (3)$$

**Microscopy**. Transgenic animals carrying MRP-1::GFP or animals stained with the MitoTracker Red CMXRos (Invitrogen™ M7512) fluorescent dye were visualized using a Leica DM4-B microscope. For the mitochondrial MRP-1::GFP colocalization experiment, MRP-1::GFP transgenic L4-stage animals were exposed to 0.5 µl of MitoTracker Red dye (1 µg/µl concentration) prepared in 1 ml of M9 buffer. The animals were incubated at a temperature of 20 °C for 5 h, followed by three washes with M9 buffer. Levamisole (0.25 mM) was used to immobilize the animals. Following acquisition, the images were rotated, cropped, and resized if necessary, using Adobe Photoshop.

**Statistics and reproducibility**. The statistical analysis was performed using Prism 9.2.0.332 (GraphPad). Data are presented as the means and standard deviations. Each experiment is repeated at least three times independently or otherwise indicated. The survival rate between groups of animals was compared using the log-rank (Mantel–Cox) test for comparison of survival curves. Differences between groups were compared using Student's t-test (two groups) or ANOVA (more than two groups), followed by suitable post hoc multiple- comparison tests. The differences were judged to be statistically significant when $P < 0.05$. 'ns' indicates nonsignificant, and '*' indicates a significant difference, as follows: *$P \leq 0.05$; **$P \leq 0.005$, ***$P \leq 0.0005$, and ****$P \leq 0.0001$.

**Reporting summary**. Further information on research design is available in the Nature Research Reporting Summary linked to this article.

## Data availability

All source data generated and/or analyzed during this study are included in this article (and its Supplementary Information and Supplementary Data 1) or are available from the corresponding authors on reasonable request.

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

## Acknowledgements
We thank the *Caenorhabditis* Genetics Center (Univ. of Minnesota) for providing the strains used in this study. Images were created using Microsoft PowerPoint and BioRender.com. We are also grateful to Aballay Lab members, Dr. Supender Kaur, Dr. Yu Sang, Dr. Benson Otarigho, and Adam Filipowicz for their valuable suggestions. This work was supported by National Institute of Health (NIH) grants GM070977 and AI156900 to A.A.

## Author contributions
J.L. and A.A. conceived and designed the experiments. J.L. performed the experiments. J.L. and A.A. analyzed the data and wrote the paper.

## Competing interests
The authors declare no competing interests.
