## [Peer Review File · Communications Biology]

Reviewers' comments:

Reviewer #1 (Remarks to the Author):

The manuscript by Lalsiamthara & Aballay proposes a new means of inter-tissue communication in *C. elegans* and its relevance in the context of innate immune surveillance. The authors propose that PA infection in *C. elegans* causes redox imbalance in intestinal cells, which leads to efflux of GSSG into the pseudocoelomic cavity through MRP-1. The extracellular GSSG was shown to be able to trigger pathogen avoidance and prime naïve *C. elegans* to induce aversive learning through neuronal NMDA class glutamate receptor, NMR-1. The paper is well written and provides interesting new findings. I would be supportive of publication provided that the authors are able to address the points listed below.

Major Comments

1. I did not find particularly convincing the images provided to show localization of MRP-1 at the basolateral membrane (Fig 2C-D), which is a key aspect in the model. Co-localization with an apical domain marker might be useful to demonstrate complementarity of expression domains. Also, the images provided should be more annotated by labeling the specific cells/membranes of interest.
2. Many readers will not be familiar with how the glutathione sensor works. The authors need to elaborate more on this point in the text and provide some positive control to justify the validity of the approach.
3. There is no attempt in the current version to back up key findings with genetics. Would intestine-specific RNAi of *gsr-1* phenocopy accumulation of GSSG similar to loss of *mrp-1*; and consequently, would this make worms hypersusceptible to PA?
4. Is the infection-induced redox imbalance leading to GSSG efflux and consequent avoidance behavior specific to PA or any intestinal infection in *C. elegans* would cause this? The authors have access to other bacterial pathogens so it would be interesting to know whether they have attempted these experiments and how generalizable the findings can be.
5. In lines 220-231, the authors present an interesting idea about context dependent role of alarmins. This should be discussed with appropriate examples (if any) in the Discussion section. How do they envisage this in the case of PA and GSSG? Does GSSG supplementation on dead/inactive PA trigger any avoidance behavior? How do the authors exclude a dose dependent effect ie. Higher dose of GSSG supplementation may be required in *E. coli*?
6. The authors show that *nrm-1* is not required for mediating pathogen avoidance but is required for GSSG-mediated aversive learning. The transition from avoidance to learning will not be very clear to the reader. Have the authors tested the PA susceptibility of *nrm-1* mutants? If GSSG release is dependent on *mrp-1*, what do the authors predict about the lawn occupancy and PA-susceptibility of *nrm-1*;*mrp-1* double mutants?

Minor Comments

1. The statistical tests for all survival curves must be included in the figures.
2. Fig S2A, the x-axis shows time and log2 in brackets. Is this a mistake?
3. Fig 4A and 4F show exactly the same thing? The authors may wish to improve data presentation to avoid having to show the same panel twice.
4. Fig 5B is low resolution on the version I was asked to review.

Reviewer #2 (Remarks to the Author):

In "The gut efflux pump MRP-1 exports oxidized glutathione as a danger signal that stimulates behavioral immunity and aversive learning" the authors show that multidrug resistance-associated protein-1 (MRP-1) functions in the basolateral membrane of intestinal cells to transport oxidized glutathione (GSSG) to control both molecular and behavioral immunity in *Caenorhabditis elegans*. The substrate, GSSG, is shown to require neural NMDA class glutamate receptor-1 (NMR-1) to induce the aversive learning behavior. The identification of GSSG as part of the signal influencing immunity and behavior during pathogen challenge is a new and important finding. Overall, this is a very interesting, informative, and convincing study that is well

carried out with appropriate controls and statistical analyses. However, it could be improved by considering the following suggestions and requests for clarification.

1) GSSG is clearly required but not sufficient for aversive learning. Any thoughts on what the other signal(s) are?

2) There is a significant body of literature suggesting that oxidants are formed in *C. elegans* during infection from a variety of cellular sources including the mitochondria and NADPH oxidases. More background on this information would nicely premise this study's finding that an oxidized molecule, GSSG, plays an important signaling role during infection.

3) If MRP-1 pumps GSSG from the intestinal cells to the pseudocoelom, then why does the overall signal from the sensor in an *mrp-1* background increase? Isn't the signal more in one part of the body than the other? Is the signal only seen when intracellular? Does release of GSSG into the pseudocoelom help get rid of it somehow?

4) NMR-1 was identified previously as being involved in pathogen aversion. When lost, does it, like MRP-1, have a pathogen sensitivity phenotype unrelated to pathogen aversion or are its effects on immunity solely through its effects on behavior?

5) Supplementary Figure 5 lacks key information. The genes knocked down are related to DMP, but what they are is not described in the figure legend or the main text. Also, some of the genes exhibited a significant difference, albeit less GSSG signal. Any thoughts about how loss of certain genes might cause this result?

6) Supplementary Figure 2A – log2 should not be in the x-axis.

Response to all reviewers

We would like to thank the reviewers for their interest in our studies and their thorough review. We have responded by performing several additional experiments and by making appropriate changes to the paper. The new experiments we have added to the manuscript to address the reviewers' comments are shown in new supplementary Fig. 3, 5, 7, 10 & 11. We are grateful for the critiques, which have made the paper much stronger, and hope the reviewers find our responses satisfactory. Changes are highlighted in gray.

Reviewer #1:

COMMENT: *The manuscript by Lalsiamthara & Aballay proposes a new means of inter-tissue communication in C. elegans and its relevance in the context of innate immune surveillance. The authors propose that PA infection in C. elegans causes redox imbalance in intestinal cells, which leads to efflux of GSSG into the pseudocoelomic cavity through MRP-1. The extracellular GSSG was shown to be able to trigger pathogen avoidance and prime naïve C. elegans to induce aversive learning through neuronal NMDA class glutamate receptor, NMR-1. The paper is well written and provides interesting new findings. I would be supportive of publication provided that the authors are able to address the points listed below.*

RESPONSE: We thank the reviewer for an excellent summary and the support of our study. We believe we have addressed all the deficiencies raised to strengthen the manuscript.

Major Comments

COMMENT: *1. I did not find particularly convincing the images provided to show localization of MRP-1 at the basolateral membrane (Fig 2C-D), which is a key aspect in the model. Co-localization with an apical domain marker might be useful to demonstrate complementarity of expression domains. Also, the images provided should be more annotated by labeling the specific cells/membranes of interest.*

RESPONSE: We performed additional experiments to further investigate the subcellular localization of MRP-1. We used the VHA-6 protein that is an apical domain marker [Allman, Erik et al. "Loss of the apical V-ATPase α -subunit VHA-6 prevents acidification of the intestinal lumen during a rhythmic behavior in *C. elegans*." American journal of physiology. Cell physiology vol. 297,5 (2009): C1071-81.]. We generated a transgenic animal that has the apical and basal region of the intestine labeled with red (VHA-6::dsRED) and green (MRP-1::GFP) fluorescence protein markers, respectively. We observed that the green and red fluorescent-tagged MRP-1 and VHA-6 proteins did not co-localize in the intestine (**new Supplementary Fig. 5**), thus confirming that the MRP-1 localizes mainly at the basolateral domain of the intestine.

We have enlarged the original **Fig. 2C**. And also, as requested by the reviewer, we have included new annotations to label the cells and membrane of interest on original **Fig. 2D**.

COMMENT: *2. Many readers will not be familiar with how the glutathione sensor works. The authors need to elaborate more on this point in the text and provide some positive control to justify the validity of the approach.*

RESPONSE: We thank the reviewer for the suggestion that improves the readability of the manuscript. We have included new sentences that would help explain the mechanism and properties of the glutathione sensor, and also the relevant characteristics of the *C. elegans* JV2 sensor strain.

The revised manuscript reads: “Genetically encoded reduction-oxidation sensitive probes such as the redox-sensitive GFP (roGFP) and its variant roGFP2 allow real-time visualization of the oxidation state of the sensor. The roGFPs have two fluorescence excitation maxima at about 400 and 490 nm, and display rapid and reversible ratiometric changes in fluorescence in response to changes in ambient redox potential⁴⁷. The fusion of glutaredoxin (Grx) protein with roGFP2 further increases the specificity for glutathione, enhances the kinetics of equilibration between the glutathione and roGFP2-Grx redox couples, and effectively allows measurement of the glutathione redox potential⁴⁸. The transgenic sensor strain JV2 was generated by expressing Grx1-roGFP2 under the large ribosomal subunit 17 (rpl-17) gene promoter⁴⁶. Thus, the sensor protein is expressed globally and constitutively within the cytosol of tissues of the animal.

To determine whether the roGFP-sensors would respond to changes in the systemic glutathione levels, we soaked the animals in a cell-permeable GSSG methyl ester to mimic GSSG accumulation inside the body of the animals. By measuring the signal intensity of the fluorescence emitted by the Grx1-roGFP2 sensor, we quantified the relative levels of GSSG and GSH in the treated and untreated animals (**Supplementary Fig. 7**). This process has helped us to validate and standardize our approach for detecting the changes in the redox balance of infected animals.”

We have also included in the Methods section the standardization using permeable GSSG as a positive control (**new supplementary Fig. 7**). The revised Methods section reads: “For the standardization of relative GSH-GSSG quantification assay, young gravid adult animals were transferred to a snap cap-tube containing S-basal medium and 5% saturated *E. coli* OP50 broth culture, which was supplemented with 25 mM (final concentration) permeable methyl-GSSG, or a snap cap-tube containing S-basal medium without the supplementation. The animals were incubated in a volume of 100 µl medium at 25°C for 4-5 h, with frequent tapping agitation. The animals were washed thrice with M9 buffer, and the GSH and GSSG profiles of the animals were determined using a COPAS Biosort flow cytometer (Union Biometrica Inc.). The animals were gated based on the extinction and time-of-flight (TOF) parameters. To measure GSSG levels, a 405 nm excitation laser was used along with a 405 nm bandpass excitation filter with a neutral density (ND) of 0.6. Using this configuration, the instrument settings were optimized, and the emission data were collected via a 512/25 nm bandpass emission filter (green channel). The biosorter acquisition parameters were set as follows: Extinction Gains- 100.0; 512/25 Green Gains- 6.0, PMT Volts- 700; 405 nm Laser Power- 100 mW. After the measurement of the GSSG profiles, the animals were harvested again through bulk sorting for the subsequent determination of the GSH profiles. To measure GSH levels, a 488 nm excitation laser was used along with a 488 nm bandpass excitation filter with an ND of 1.5. Using this configuration, the instrument settings were optimized, and the emission data were collected using a 512/25 nm bandpass emission filter (green channel). The biosorter acquisition parameters were set as follows: Extinction Gains- 2.0; 512/25 Green Gains- 2.0, PMT Volts 390; 488 nm Laser Power- 50 mW. For the measurement of GSH or GSSG profiles of animals exposed to food or pathogen, young gravid adult animals were harvested in M9 buffer and transferred to a 6 cm NGM plate containing a 16 h lawn of *P. aeruginosa* (PA14) or control *E. coli* (OP50). After 4–5 h of incubation at 25°C, the animals were collected and washed thrice with M9 buffer to remove bacteria and debris. The GSH or GSSG profiles of the animals were determined using a COPAS Biosort flow cytometer. The acquired data was exported into Excel files and subjected to analysis using Prism 8 (GraphPad).”

COMMENT: 3. There is no attempt in the current version to back up key findings with genetics. Would intestine-specific RNAi of *gsr-1* phenocopy accumulation of GSSG similar to loss of *mrp-1*; and consequently, would this make worms hypersusceptible to PA?

RESPONSE: We disagree with the reviewer on this issue as the central theme of our work is the role of the gut efflux pump MRP-1 in the control of behavioral immunity. We have demonstrated this by using mutant animals, transgenic animals that show the tissue-specific expression of MRP-1 and its cellular membrane localization, and using a tissue-specific rescue strain.

While we believe that further studies concerning the genetic control of GSSG are outside the scope of our work, we did try to address the reviewer comment. Unfortunately, we faced technical difficulties as most of the relevant genes are crucial for embryonic development. The

genes of interest that cause embryonic lethality include: i) the genes encoding the enzymes for bio-synthesis of glutathione, *gcs-1* (**gamma GlutamylCysteine Synthetase**), and *gss-1* (**Glutathione Synthetase**) [Romero-Aristizabal, Catalina et al. “Regulated spatial organization and sensitivity of cytosolic protein oxidation in *Caenorhabditis elegans*.” *Nature communications* vol. 5 5020. 29 Sep. 2014.; *gss-1(tm672)* deletion mutant has been reported as lethal/sterile by the NBRP *C. elegans* Gene Knockout Consortium (<http://www.shigen.nig.ac.jp/c.elegans/>); ii) the genes required for the interconversion of glutathione species, *gpx-1* (**Glutathione PeroXidase**) and *gsr-1* (**Glutathione diSulfide Reductase**) [Harris, T.W., Arnaboldi, V., Cain, S., Chan, J., Chen, W.J., Cho, J., Davis, P., Gao, S., Grove, C.A., Kishore, R., et al. (2020). WormBase: A modern Model Organism Information Resource. *Nucleic Acids Res.* 48, D762–D767; Mora-Lorca, José Antonio et al. “Glutathione reductase *gsr-1* is an essential gene required for *Caenorhabditis elegans* early embryonic development.” *Free radical biology & medicine* vol. 96 (2016): 446-61].

To address the reviewer’s comment, we performed RNAi mediated knock-down of the *gsr-1* mRNA in the intestine of L4-stage animals (MGH171 intestine-specific RNAi strain). Under these conditions, we did not observe any difference in pathogen susceptibility (**please see Fig. 1 below**).

Fig. 1. Representative survival plot of intestinal specific RNAi strain (MGH171) exposed to a partial lawn of *P. aeruginosa*, following *mrp-1*, *gsr-1* and vector only RNAi at L4-stage and induced for 48 h. Assays were performed at 25°C, with 40–50 animals/assay. (Curve comparison, vector control versus *mrp-1*, significant ****, P value <0.0001; vector control versus *gsr-1*, non-significant, P value 0.7176 via Log-rank (Mantel-Cox) test).

The lack of phenotype is not surprising as it has been reported that the knockdown of *gsr-1* hardly affected total glutathione levels nor reduced glutathione/glutathione disulfide (GSH/GSSG) ratio in *C. elegans* under normal laboratory conditions [Lüersen, Kai et al. “The glutathione reductase GSR-1 determines stress tolerance and longevity in *Caenorhabditis elegans*.” *PloS one* vol. 8,4 e60731. 8 Apr. 2013]. Because of the lack of a phenotype of *gsr-1* RNAi animals and the published lack of effect of *gsr-1* RNAi on total glutathione levels, we decided not to include this result in the manuscript.

COMMENT: 4. *Is the infection-induced redox imbalance leading to GSSG efflux and consequent avoidance behavior specific to PA or any intestinal infection in C. elegans would cause this? The authors have access to other bacterial pathogens so it would be interesting to know whether they have attempted these experiments and how generalizable the findings can be.*

RESPONSE: We thank the reviewer for suggesting the experiments. To test whether the observed behavior is specific to PA, we performed additional experiments. We found that the *mrp-1* mediated behavioral control of immunity may be specific to *P. aeruginosa* infection. The revised manuscript reads: “*We also examined the avoidance behavior of mrp-1(pk89) mutant animals towards other bacterial species known to cause avoidance of C. elegans. We found that the lawn occupancy of mrp-1(pk89) mutant animals was comparable to that of wild-type animals on lawns of E. faecalis and S. aureus (Supplementary Fig. 3). This finding suggests that mrp-1-mediated behavioral control of pathogen avoidance is not universal and may be specific to P. aeruginosa.*”.

COMMENT: 5. *In lines 220-231, the authors present an interesting idea about context dependent role of alarmins. This should be discussed with appropriate examples (if any) in the Discussion section. How do they envisage this in the case of PA and GSSG ? Does GSSG supplementation on dead/inactive PA trigger any avoidance behavior? How do the authors exclude a dose dependent effect ie. Higher dose of GSSG supplementation may be required in E. coli?*

RESPONSE: We have not been able to identify examples worth discussion in the context of our study. We have performed the suggested experiments and discussed them in the revised manuscript.

- *Does GSSG supplementation on dead/inactive PA trigger any avoidance behavior?*

We performed the suggested experiment and found that live PA is required for the effect of GSSG on avoidance (**new supplementary Fig. 10**). The revised manuscript reads: “*We observed that the addition of GSSG to the lawn induced enhanced pathogen avoidance in the presence of live but not killed P. aeruginosa (Fig. 4A and Supplementary Fig. 10).*”

- *How do the authors exclude a dose dependent effect ie. Higher dose of GSSG supplementation may be required in E. coli?*

We performed the experiment and found that higher concentrations of GSSG do not elicit avoidance in the presence of *E. coli* (**new supplementary Fig. 11**). The revised manuscript reads: “*We next explored the possibility that the GSSG molecule functions as an alarmin that requires the presence of pathogenic bacteria to promote lawn-leaving behavior in animals. We determined the occupancy of animals on lawns of nonpathogenic E. coli supplemented with GSSG. In contrast to the GSSG-enhanced P. aeruginosa avoidance observed (Fig. 4A), we observed no GSSG-mediated lawn-leaving behavior upon interaction with nonpathogenic E. coli (Fig. 4E-G). Even higher GSSG concentrations than those capable of inducing avoidance of P. aeruginosa also failed to induce avoidance of E. coli (Supplementary Fig. 11), suggesting that the ability of GSSG to induce aversive behavior was pathogen dependent.*”

COMMENT: 6. *The authors show that nrm-1 is not required for mediating pathogen avoidance but is required for GSSG-mediated aversive learning. The transition from avoidance to learning will not be very clear to the reader. Have the authors tested the PA susceptibility of nrm-1 mutants? If GSSG release is dependent on mrp-1, what do the authors predict about the lawn occupancy and PA-susceptibility of nrm-1;mrp-1 double mutants?*

RESPONSE: We believe that we have sufficiently explained the rationale for studying aversive learning sufficiently clearly at the beginning of the section entitled “The NMR-1 receptor is required for GSSG-mediated adult learned aversion.” Also, in the revised manuscript, we have made clearer that for learned pathogen aversion: “*These results suggest that NMR-1 is not the receptor that is involved in the lawn occupancy of naïve animals; however, it is a receptor required for learned pathogen aversion as well as GSSG-mediated adult learned pathogen aversion*”.

We performed the requested experiment and found that, as expected, *nmr-1* animals are not susceptible to *P. aeruginosa*. We believe that the inclusion of these results would be a distraction from the main message of the paper.

Fig. 2. Representative survival plot of wild-type, *mrp-1(pk89)*, *nmr-1(ak4)*, and *mrp-1;nmr-1* mutant animals exposed to a partial lawn of *P. aeruginosa*. (Curve comparison, wild type versus *mrp-1*, ****, P value <0.0001; wild type versus *nmr-1*, ns, P value = 0.4056; wild type versus *mrp-1;nmr-1*, ****, P value <0.0001; *mrp-1* versus *nmr-1*, ****, P value <0.0001; *mrp-1* versus *mrp-1;nmr-1*, ns, P value = 0.5617. “*” indicates significant, ns-non-significant, P value determined via Log-rank (Mantel-Cox) test). Assays were performed at 25°C, with 40–50 animals/assay.

Because *nmr-1* is required for adult-learned aversion and *mrp-1* is required for naïve aversion, we do not expect the behavior of the double mutant to be different than that of the single mutants.

Minor Comments

COMMENT: 1. The statistical tests for all survival curves must be included in the figures.

RESPONSE: We have added the statistical test for the survival curve in the revised Methods section. The log-rank (Mantel-Cox) test was used to calculate the statistical significance between survival curves. We have added the P values for all relevant comparisons of survival curves in the figure legends of the revised manuscript.

COMMENT: 2. Fig S2A, the x-axis shows time and log2 in brackets. Is this a mistake?

RESPONSE: We apologize for this error. This has been corrected. We have removed the ‘log2’ label from the **supplementary Fig. 2A** in the revised manuscript.

COMMENT: 3. Fig 4A and 4F show exactly the same thing? The authors may wish to improve data presentation to avoid having to show the same panel twice.

RESPONSE: We made the suggested change. We have removed the repetitive **Fig. 4F**, and changed the figure labels accordingly in the revised manuscript.

COMMENT: 4. Fig 5B is low resolution on the version I was asked to review.

RESPONSE: We have added new representative images for **Fig.5B** that may help better visualization.

Reviewer #2

COMMENT: In “The gut efflux pump MRP-1 exports oxidized glutathione as a danger signal that stimulates behavioral immunity and aversive learning” the authors show that multidrug resistance-associated protein-1 (MRP-1) functions in the basolateral membrane of intestinal cells to transport oxidized glutathione (GSSG) to control both molecular and behavioral immunity in *Caenorhabditis elegans*. The substrate, GSSG, is shown to require neural NMDA class glutamate receptor-1 (NMR-1) to induce the aversive learning behavior. The identification of GSSG as part of the signal influencing immunity and behavior during pathogen challenge is a new and important finding. Overall, this is a very interesting, informative, and convincing study that is well carried out with appropriate controls and statistical analyses. However, it could be improved by considering the following suggestions and requests for clarification.

RESPONSE: We thank the reviewer for an excellent summary of the work and for the support of the study.

COMMENT: 1) GSSG is clearly required but not sufficient for aversive learning. Any thoughts on what the other signal(s) are?

RESPONSE: Our results actually show that GSSG is indeed sufficient for aversive learning (**Fig. 5C and D**). We believe the reviewer probably meant that GSSG is not sufficient for naïve aversion of *E. coli*. New experiments suggest that live *P. aeruginosa* is required for the effect of GSSG (**new supplementary Fig. 10**). The revised manuscript reads: “We observed that the addition of GSSG to the lawn induced enhanced pathogen avoidance in the presence of live but not killed *P. aeruginosa* (**Fig. 4A and Supplementary Fig. 10**). Supplementation of GSSG was also capable of inducing pathogen avoidance in *mrp-1(pk89)* mutant animals (**Fig. 4B**).”

COMMENT: 2) There is a significant body of literature suggesting that oxidants are formed in *C. elegans* during infection from a variety of cellular sources including the mitochondria and NADPH oxidases. More background on this information would nicely premise this study’s finding that an oxidized molecule, GSSG, plays an important signaling role during infection.

RESPONSE: We thank the reviewer for this suggestion. Because the focus of our work is the gut efflux pump MRP-1, we focused our discussion on ATP-binding cassette (ABC) transporters and a putative substrate of MRP-1, GSSG. We have revised the discussion to include information regarding NADPH oxidases and ROS. The revised manuscript reads: “Several lines of evidence indicate that redox-signaling is one of the many regulators of *C. elegans* behavior in response to infectious pathogens^{65,66}. *C. elegans*, in response to microbial infection, produces several reactive oxygen species (ROS)^{67,68}. The dual oxidase *Ce-Duox1/BLI-3*, a member of the NADPH-oxidase family, is responsible for ROS generation in *C. elegans*^{67,69}. It has been reported that elevated ROS may result in oxidative stress leading to cellular injury and damage. However, to mount appropriate counteractive and restorative responses, the cell requires a functioning

redox signaling, which is regulated by the different ROS species⁷⁰. Byproducts of redox-imbalance, such as accumulation of GSSG molecules, in combination with pathogens, may serve as alarmins that alert the animal of an impending danger.”

COMMENT: 3) *If MRP-1 pumps GSSG from the intestinal cells to the pseudocoelom, then why does the overall signal from the sensor in an mrp-1 background increase? Isn't the signal more in one part of the body than the other? Is the signal only seen when intracellular? Does release of GSSG into the pseudocoelom help get rid of it somehow?*

RESPONSE: The overall signal from the sensor in an *mrp-1* background increases because the loss of efflux pump resulted in the accumulation of GSSG within the cells, and the accumulated GSSG interacts readily with the roGFP sensor proteins, which are solely expressed in the cytosol of the cells. This increased interaction causes conformational changes in roGFP structure and results in increased signal intensity during the quantification.

- *Isn't the signal more in one part of the body than the other?*

The roGFP sensor is expressed constitutively and globally across various tissues under the *rpl-17* gene promoter in the reporter strain.

- *Is the signal only seen when intracellular?*

There is no export signal that would export GFP outside the cells.

- *Does release of GSSG into the pseudocoelom help get rid of it somehow?*

As stated above, we believe that in the absence of MRP-1 GSSG accumulates inside the cells, increasing the signal. We believe that in wild-type animals, MRP-1 exports GSSG into the pseudocoelom, thus reducing the signal.

We have revised the manuscript to better explain the sensor strain: “*Genetically encoded reduction-oxidation sensitive probes such as the redox-sensitive GFP (roGFP) and its variant roGFP2 allow real-time visualization of the oxidation state of the sensor. The roGFPs have two fluorescence excitation maxima at about 400 and 490 nm, and display rapid and reversible ratiometric changes in fluorescence in response to changes in ambient redox potential⁴⁷. The fusion of glutaredoxin (Grx) protein with roGFP2 further increases the specificity for glutathione, enhances the kinetics of equilibration between the glutathione and roGFP2-Grx redox couples, and effectively allows measurement of the glutathione redox potential⁴⁸. The transgenic sensor strain JV2 was generated by expressing Grx1-roGFP2 under the large ribosomal subunit 17 (*rpl-17*) gene promoter⁴⁶. Thus, the sensor protein is expressed globally and constitutively within the cytosol of tissues of the animal. To determine whether the roGFP-sensors would respond to changes in the systemic glutathione levels, we soaked the animals in a cell-permeable GSSG methyl ester to mimic GSSG accumulation inside the body of the animals. By measuring the signal intensity of the fluorescence emitted by the Grx1-roGFP2 sensor, we quantified the relative levels of GSSG and GSH in the treated and untreated animals (Supplementary Fig. 7). This process has helped us to validate and standardize our approach for detecting the changes in the redox balance of infected animals.*”

COMMENT: 4) *NMR-1 was identified previously as being involved in pathogen aversion. When lost, does it, like MRP-1, have a pathogen sensitivity phenotype unrelated to pathogen aversion or are its effects on immunity solely through its effects on behavior?*

RESPONSE: The reviewer has raised an interesting question. Because NMR-1 is only required for learned-pathogen avoidance (i.e., it is not required for avoidance of naïve animals), we do not expect to be involved in pathogen sensitivity. We performed this experiment (please see Fig. 2

below), but we believe that the inclusion of these results would be a distraction from the main message of the paper.

Fig. 2. Representative survival plot of wild-type, *mrp-1(pk89)*, *nmr-1(ak4)*, and *mrp-1;nmr-1* mutant animals exposed to a partial lawn of *P. aeruginosa*. (Curve comparison, wild type versus *mrp-1*, ****, P value <0.0001; wild type versus *nmr-1*, ns, P value = 0.4056; wild type versus *mrp-1;nmr-1*, ****, P value <0.0001; *mrp-1* versus *nmr-1*, ****, P value <0.0001; *mrp-1* versus *mrp-1;nmr-1*, ns, P value = 0.5617. “*” indicates significant, ns-non-significant, P value determined via Log-rank (Mantel-Cox) test). Assays were performed at 25°C, with 40–50 animals/assay.

COMMENT: 5) *Supplementary Figure 5* lacks key information. The genes knocked down are related to DMP, but what they are is not described in the figure legend or the main text. Also, some of the genes exhibited a significant difference, albeit less GSSG signal. Any thoughts about how loss of certain genes might cause this result?

RESPONSE: We apologize for this shortcoming. We mentioned the genes used in the study and included relevant references to help further reading in the revised manuscript as follows “The genes *aex-5*, *f1r-1*, *nhx-2*, and *pbo-1* were selected for this study as their inactivation is known to lead to defects in the DMP, which results in the bloating and distention of the intestinal lumen by the accumulation of bacteria^{50–52}.”

Because inhibition of those genes is known to cause pathogen avoidance, we wanted to study whether the mechanism by which they may elicit avoidance was due to higher levels of GSSG. **Supplementary Fig. 8** shows that that is not the case. We do not know why those animals have lower GSSG levels.

COMMENT: 6) *Supplementary Figure 2A* – log2 should not be in the x-axis.

RESPONSE: We apologize for this error. This has been corrected. We have removed the ‘log2’ label from the **Supplementary Fig. 2A** in the revised manuscript.

REVIEWERS' COMMENTS:

Reviewer #1 (Remarks to the Author):

The authors have satisfactorily addressed all comments. I believe their interesting manuscript has further improved and I am very supportive of publication of this current version.

Reviewer #2 (Remarks to the Author):

The authors have very well addressed my initial comments and I have no remaining concerns.